



# Accelerating models for multiphase chemical kinetics through machine learning with polynomial chaos expansion and neural networks

Thomas Berkemeier[1], Matteo Krüger[1,†], Aryeh Feinberg[2,3,4,5,†], Marcel Müller[2,†], Ulrich Pöschl[1], and Ulrich K. Krieger[2]

[1]Max Planck Institute for Chemistry, Hahn-Meitner-Weg 1, 55128 Mainz, Germany
[2]Institute for Atmospheric and Climate Science, ETH Zürich, 8092 Zürich, Switzerland
[3]Institute of Biogeochemistry and Pollutant Dynamics, ETH Zürich, 8092 Zürich, Switzerland
[4]Eawag, Swiss Federal Institute of Aquatic Science and Technology, 8600 Dübendorf, Switzerland
[5]currently at Institute for Data, Systems, and Society, Massachusetts Institute of Technology, 02142 Cambridge, MA, USA.
[†]These authors contributed equally to this work.

**Correspondence:** Thomas Berkemeier (t.berkemeier@mpic.de)

**Abstract.** The heterogeneous chemistry of atmospheric aerosols involves multiphase chemical kinetics that can be described by kinetic multi-layer models (KM) explicitly resolving mass transport and chemical reaction. However, KM are computationally too expensive to be used as sub-modules in large-scale atmospheric models, and the computational costs also limit their utility in inverse modelling approaches commonly used to infer aerosol kinetic parameters from laboratory studies. In this study, we show how machine learning methods can generate inexpensive surrogate models based on the kinetic multi-layer model of aerosol surface and bulk chemistry (KM-SUB). We apply and compare two common and openly available methods for the generation of surrogate models, polynomial chaos expansion (PCE) with UQLab and neural networks (NN) through the Python package Keras. We show that the PCE method is well-suited to determine global sensitivity indices of the KM and demonstrate how inverse modelling applications can be enabled or accelerated with NN-suggested sampling. These qualities make them suitable supporting tools for laboratory work in the interpretation of data and design of future experiments. Overall, the KM surrogate models investigated in this study are fast, accurate, and robust, which suggests their applicability as sub-modules in large-scale atmospheric models.

## 1 Introduction

An accurate description of the heterogeneous chemistry of atmospheric particles requires explicit coupling of mass transport with chemical reactions (Pöschl et al., 2007; Kolb et al., 2010; Shiraiwa et al., 2014). Especially for particles containing secondary organic matter, field and laboratory experiments during the last decade showed severe transport limitations that affect chemical reactivity (Shiraiwa et al., 2011; Kuwata and Martin, 2012; Berkemeier et al., 2016). While the elementary processes are well understood, kinetic multi-layer models (KM) describing mass transport and chemical reactions at the gas-particle interface and throughout the particle bulk are computationally expensive due to the need of spatial resolution within



the particles (Pöschl et al., 2007; Shiraiwa et al., 2012; Roldin et al., 2014; Berkemeier et al., 2017; Semeniuk and Dastoor, 2020; Dou et al., 2021). For the use in global or regional models, the KM would have to be evaluated for every grid cell, time step, and particle class (size/composition). This computational volume makes the application of KM extremely costly, if not outright impossible.

A second complicating factor for KM is the multitude of chemical and physical input parameters such as transport parameters or chemical reaction rate coefficients, which are often poorly constrained or unknown. Thus, in a laboratory setting, KM are often used in an inverse modelling approach, in which model parameters are deduced or constrained with experimental data using global optimization (Berkemeier et al., 2017; Tikkanen et al., 2019; Berkemeier et al., 2021; Wei et al., 2021; Milsom et al., 2022). However, due to the inherently coupled nature of the underlying physical and chemical processes, input parameters are often ill-constrained, i.e. their numerical value cannot be uniquely determined (Berkemeier et al., 2017). This is particularly problematic when extrapolating the KM to conditions outside the calibration range where the calculation outcome can depend strongly on previously insensitive and thus unconstrained parameters (or combinations of parameters). Fit ensembles, i.e. arrays of multiple solutions from repeated execution of a global optimization algorithm, can be utilized to propagate the uncertainty of the global fit to conditions outside the calibration range (Berkemeier et al., 2021). Solving the inverse problem is a complex task that becomes computationally more expensive with increasing number of uncertain model input parameters, often requiring $>10^5$ model simulations (Xu et al., 2018). In some cases, this can be prohibitively expensive to do with a full model and the problem is exacerbated when acquiring or evaluating fit ensembles.

Computationally inexpensive surrogate models can replace KM in specialized tasks and help solving the issue of computational cost. These surrogate models are trained on a data set consisting of a wide range of kinetic input parameters and the associated calculated outputs until they reproduce the KM output with the desired accuracy. Surrogate-based optimization methods are an active field of research (Booker et al., 1999; Vu et al., 2017; Xu et al., 2018). Some studies use an iterative approach, wherein the surrogate model is used to constrain the likely parameter space and the full model is run within this likely parameter space to refine the surrogate model. Here, we illustrate the generation of surrogate models by introducing two suitable machine learning methods, namely artificial neural networks (NN) through the Python package Keras (Gulli and Pal, 2017) and polynomial chaos expansion (PCE) with UQLab (Marelli and Sudret, 2014).

Artificial NN represent a group of common machine learning algorithms. Their functionality is inspired by biological brains, where complex computational processes are based on comparably simple interactions of large numbers of interconnected nodes, or neurons (Kröse and van der Smagt, 1996). Neural networks are commonly organized in layers, where an individual neuron obtains signals from neurons in the previous layer and maps them to a single new signal that is passed to neurons of the following layer (Almeida, 2001; Popescu et al., 2009). By systematic variation of the numerical weights of individual neuron operations, the so-called training, a NN can increase its predictive accuracy. The exact mathematical operations that are performed by neurons in specific layers and the arrangement of such layers (architecture of the NN) are determined by so-called hyperparameters. Hyperparameters can be adapted to obtain a NN that is specialized on a specific task, input data structure or output type (Bishop, 1994; Sadeeq and Abdulazeez, 2020).



In the atmospheric sciences, NN are used for air quality prediction, function approximation, and pattern recognition tasks
(Gardner and Dorling, 1998), but their application as surrogate models for computationally expensive KM is less well re-
searched. Recently, popular applications of machine learning in atmospheric chemistry and physics include quantitative structure-
activity relationship (QSAR) models that map molecular structures to compound properties as an alternative to time-consuming
laboratory experiments or quantum mechanical calculations (Lu et al., 2021; Lumiaro et al., 2021; Galeazzo and Shiraiwa,
2022; Krüger et al., 2022; Xia et al., 2022). Holeňa et al. (2010) used surrogate models in computationally costly evolutionary
optimization and successfully enhanced this approach with the application of NN. Tripathy and Bilionis (2018) used a NN to
create surrogate models for expensive high dimensional uncertainty quantification. Other recent applications of NN as surro-
gate models address chemical and process engineering (Cavalcanti et al., 2021; Esche et al., 2022) or materials science (Allotey
et al., 2021).

The second method applied in this work is polynomial chaos expansion (PCE), a method commonly used for uncertainty
quantification (Sudret, 2008). In the PCE approach, the full model is represented as an infinite series of suitably-built, multi-
variate, and orthonormal polynomial functions (Marelli and Sudret, 2014). Surrogate models using PCE methods have been
developed mainly within engineering fields (Ghanem and Spanos, 2003; Sudret, 2008). Several recent environmental chem-
istry investigations have applied PCE surrogate modelling, particularly because of its suitability for global sensitivity analysis
problems (Thackray et al., 2015; Feinberg et al., 2020). The goal of global sensitivity analysis is to apportion the uncertainty in
model output into contributions from the uncertainties of different model input variables, additionally considering interacting
effects between input parameter uncertainties (Saltelli et al., 2008). The results from the sensitivity analysis indicate which
are the most influential input parameters that should be further constrained and may therefore be a useful tool in designing or
prioritizing laboratory experiments.

## 2 Methods

The surrogate modelling workflow employed in this study is shown in Fig. 1. To acquire a fast–computing surrogate model
for the computationally expensive KM, training data are first acquired by sampling outputs of the full model from the possible
model parameter space. The surrogate models are trained with Keras and UQLab on this data and are validated by comparison
with a test data set of full model output.

### 2.1 Kinetic multi-layer model KM-SUB

In this study, we employ the kinetic multi-layer model of aerosol surface and bulk chemistry (KM-SUB, Shiraiwa et al., 2010),
but the statistical methods could be used with any process model. KM-SUB describes mass transport and chemical reaction at
the surface and in the bulk of aerosol particles by solving a set of ordinary differential equations. The model explicitly treats gas
diffusion, surface and bulk accommodation of gas molecules, surface-bulk exchange, and bulk diffusion as well as chemical
reaction at the surface and in the bulk of aerosol particles. For a schematic depiction of the processes and compartments of
KM-SUB, see Fig. B1.



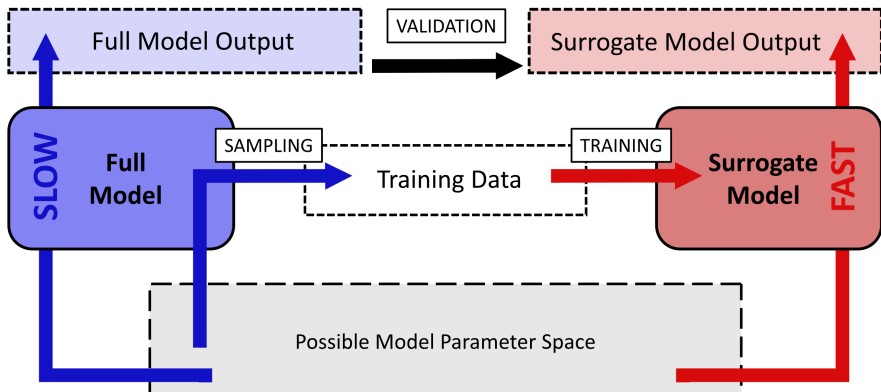

**Figure 1.** Workflow chart for the surrogate modelling process employed in this study. The possible or desired model parameter space (gray) is sampled with the slow computing full model (blue) to acquire training data consisting of model input/output pairs. Training data is used for training of a fast computing surrogate model (red). Surrogate models are validated by comparison of full model output and surrogate model output.

For the model calculations in this study, we chose a general model scenario of a single volatile reactant X (e.g. OH, $O_3$, $NO_3$) reacting with a single non-volatile reactant Y at the surface and in the bulk of the aerosol particle. The input parameters of KM-SUB resulting from this scenario include initial concentrations, reaction rate coefficients, and diffusion coefficients (Table 1). The outputs of KM-SUB are concentration profiles over space and time, but in this study, we summarized KM-SUB output as the total number of Y in a single aerosol particle at time $t$ ($N_{Y,t}$). To minimize data storage requirements, we reduce the full KM-SUB time series to three output values, the time required to reach 90 %, 50 % (i.e. the chemical half-life), and 10 % of $N_{Y,0}$ by interpolation of primary model output. The in- and outputs of KM-SUB are then log-transformed. For the NN application, all input parameters and model outputs are additionally normalized to the interval [0:1]. Outputs are normalized by dividing by the longest time recorded to reach 10 % of $N_{Y,0}$.

For each input parameter of KM-SUB, individual parameter boundaries are defined that represent a wide array of reactants and scenarios that can be found in either the atmosphere or in laboratory experiments (Table 1). As these ranges cover orders of magnitude, they are assumed to follow log-uniform probability distributions. The parameter space includes liquid to semisolid particles (as expressed by the reactant diffusivities) from 50 nm to 100 µm in size. Reaction rate coefficients range from reactivity close to the diffusion limit, typical for the OH radical ($1\times10^{11}$ cm$^3$ s$^{-1}$), down to reactions that are nine orders of magnitude slower and may be associated with reactions involving ozone. The volatile reactant X is given a large variability in partitioning properties (as expressed by surface accommodation coefficient $\alpha_{s,0}$ and desorption lifetime $\tau_d$) and solubility properties (as expressed by the Henry's law coefficient), each varying over several orders of magnitude. The initial concentration of non-volatile reactant Y ranges from $10^{19}$ cm$^{-3}$ to $2\times10^{21}$ cm$^{-3}$, which for an organic substance with molar mass of 250 g mol$^{-1}$ corresponds roughly to a molar fraction from 0.5 % to pure particles. The concentration of X in the gas phase is held constant over a simulation and varied between simulations from a few parts per billion ($10^{11}$ molecules cm$^{-3}$) to about 200



**Table 1.** KM-SUB input parameters with lower and upper boundaries and fit parameters to the laboratory data set.

| Parameter | Lower boundary | Upper boundary | Description |
|---|---|---|---|
| $k_{\mathrm{SLR}}$ | $1.0 \times 10^{-15}$ | $1.0 \times 10^{-8}$ | Rate coefficient of X+Y surface reaction ($\mathrm{cm^2\ s^{-1}}$) |
| $k_{\mathrm{BR}}$ | $1.0 \times 10^{-20}$ | $1.0 \times 10^{-11}$ | Rate coefficient of X+Y bulk reaction ($\mathrm{cm^3\ s^{-1}}$) |
| $D_{\mathrm{b,X}}$ | $1.0 \times 10^{-11}$ | $1.0 \times 10^{-5}$ | Bulk diffusion coefficient of X ($\mathrm{cm^2\ s^{-1}}$) |
| $D_{\mathrm{b,Y}}$ | $1.0 \times 10^{-12}$ | $1.0 \times 10^{-6}$ | Bulk diffusion coefficient of Y ($\mathrm{cm^2\ s^{-1}}$) |
| $H_{\mathrm{cp,X}}$ | $5.0 \times 10^{-6}$ | $5.0 \times 10^{-3}$ | Henry's law solubility coefficient of X ($\mathrm{mol\ cm^{-3}\ atm^{-1}}$) |
| $\tau_{\mathrm{d,X}}$ | $1.0 \times 10^{-9}$ | $1.0 \times 10^{-2}$ | Desorption lifetime of X (s) |
| $\alpha_{\mathrm{s,0,X}}$ | $1.0 \times 10^{-4}$ | $1$ | Surface accommodation coefficient of X |
| | | | on an adsorbate-free surface (unitless) |
| $r_{\mathrm{p}}$ | $2.5 \times 10^{-6}$ | $1.0 \times 10^{-3}$ | Particle radius (cm) |
| $[\mathrm{X}]_{\mathrm{g,0}}$ | $1.0 \times 10^{11}$ | $1.0 \times 10^{15}$ | Initial gas phase number concentration of X ($\mathrm{cm^{-3}}$) |
| $[\mathrm{Y}]_{\mathrm{b,0}}$ | $1.0 \times 10^{19}$ | $2.0 \times 10^{21}$ | Initial bulk number concentration of Y ($\mathrm{cm^{-3}}$) |

parts per million ($5 \times 10^{15}$ molecules $\mathrm{cm^{-3}}$). For the explicit treatment of gas diffusion, we assume a temperature of 298 K and a fixed diffusion coefficient of 0.14 $\mathrm{cm^2\ s^{-1}}$.

## 2.2 Acquisition of training data

The KM is used to generate a training data set for the surrogate models by randomly sampling parameters in log-uniform space
within their associated boundaries. The number of KM samples obtained in this study is about $4.3 \times 10^6$ and required super-computing. A random set of 1000 samples is removed from the data set and withheld from model training for the visualization and validation of fully-trained surrogate models. We refer to this set of data as "test data".

As not only the computational effort of sampling training data, but also the time required for surrogate model creation increases with the size of the training data set, the surrogate model performance is tested on different fractions of the total
training data set in order to find an optimal or sufficient computational expense for a given application (Table 2). Note that the PCE method is only applied to the first nine fractions (50 - 20,000) due to the computational expense of the method at higher training set sizes.

## 2.3 Neural network (NN)

The neural network architecture employed in this study is a multilayer perceptron (MLP), in which nodes are organized in
consecutive layers. MLP are characterized by a chosen number of so-called "hidden" layers that connect the "visible" in- and output layers. Each node in a layer is connected with each node in the previous and following layer (fully connected layers). We test MLP consisting of up to five hidden layers with variable numbers of neurons to determine a network architecture that suits the specified task. A detailed mathematical description of MLP functionality and architecture is given in Appendix A1.





The processes of hyperparameter tuning, tested ranges, and suggested values for individual hyperparameters are described
in Appendix A2. We apply 5-fold cross-validation to avoid over-fitting of the trained models during hyperparameter tuning
(Stone, 1974; Wong and Yeh, 2020).

### 2.4 Polynomial chaos expansion (PCE)

The PCE surrogate modelling approach will be briefly summarized here. For more technical descriptions the reader can refer to
Sudret (2008) and Le Gratiet et al. (2017). The principle behind PCE is that the model output $Z$ is decomposed into an infinite
series (Ghanem and Spanos, 2003):

$$Z = \sum_{\alpha \in \mathbb{N}^M} y_\alpha \psi_\alpha(X) \tag{1}$$

where $M$ is the number of model input variables, $\alpha$ is a multi-index that defines the variable components of the polynomials,
$y_\alpha$ are coefficients, and $\psi_\alpha$ are orthonormal polynomials of either one input variable (representing first-order effects) or multiple
input variables (representing interacting effects). The type of orthonormal polynomial in Eq. 1 depends on the probability
distribution of the input parameters, with uniform probability distributions being represented by Legendre polynomials and
Gaussian probability distributions by Hermite polynomials (Xiu and Karniadakis, 2002). In practice, Eq. 1 is truncated by
restricting the maximum degree of the polynomials. We calculate PCE coefficients ($y_\alpha$) using the implementation of least-
angle regression (Blatman and Sudret, 2010) from the open-source Matlab-based software UQLab (Marelli and Sudret, 2014).
This software allows degree-adaptive calculation of the PCE, meaning that PCE models can be constructed from degree 1 to a
maximum selected degree, which we set to 14. If the cross-validation error of the model does not decrease over two steps in
degree, the algorithm stops and the PCE with the lowest cross-validation error is selected. All PCE calculated for this study are
equal or below degree 7 (Table A1).

### 2.5 Global sensitivity analysis

In global sensitivity analysis, Sobol' indices describe the contribution of uncertainty from each input parameter and interactions
between input parameters (Sobol', 2001). The variance ($D$) of the model output $Z$ is decomposed into partial variances:

$$D = \text{Var}(Z) = \sum_{i=1}^{M} D_i + \sum_{1 \leq i < j \leq M} D_{ij} + \text{ higher order terms} \tag{2}$$

i.e. the sum of first-order partial variances ($D_i$), second order partial variances ($D_{ij}$), and higher order terms. Sobol' indices
($S$) are calculated by normalizing the partial variances by the total variances, e.g. $S_i = \frac{D_i}{D}$ for the first-order contribution of
$i$th input parameter and $S_{ij} = \frac{D_{ij}}{D}$ for the contribution of the interaction between the $i$th and $j$th input parameters to the model
uncertainty. In order to summarize the overall influence of a specific input parameter, including interactions, a total Sobol'
index ($S_i^T$) can be calculated:





$$S_i^T = S_i + \sum_{j \neq i}^{M} S_{ij} + \sum_{j \neq i} \sum_{\substack{k \neq i \\ k \neq j}} S_{ijk} + ... + S_{ij...M} \tag{3}$$

Given the similarities between the PCE and Sobol' decompositions, the Sobol' sensitivity indices can be calculated analytically from the PCE coefficients, rather than with Monte Carlo sampling (Sudret, 2008). This eliminates a potentially
computationally expensive step of the sensitivity analysis process with other surrogate models.

### 2.6 Acquisition of fit ensembles

With the trained NN model, we illustrate and test the application of surrogate models in inverse modelling approaches with KM-SUB. Six sets of experimental data of the well-studied oleic acid ozonolysis heterogeneous reaction system (Hearn and Smith, 2004; Ziemann, 2005; Gallimore et al., 2017; Berkemeier et al., 2021) are used to determine kinetic parameter sets
that minimize the mean squared (absolute) logarithmic error (MSLE) between model and experiments. More details about the specific optimization problem can be found in Appendix B.

$$\text{MSLE} = \frac{1}{N} \sum_{i=1}^{N} \frac{1}{n} \sum_{j=1}^{n} (\log_{10}(z_{ij}) - \log_{10}(y_{ij}))^2 \tag{4}$$

where $N$ is the number of experimental data sets, $n$ the number of data points in each set, $z_{ij}$ the model output, and $y_{ij}$ the value for experiment $i$ and data point $j$. As this optimization problem does not offer a unique solution (Berkemeier et al., 2021),
the aim is not to find a best-fitting parameter set, but rather a fit ensemble, i.e. an array of parameter sets that all yield a sufficient agreement of the associated KM-SUB outputs with the experimental data. The fit ensemble then represents not only the ranges to which kinetic input parameters could be constrained, but is also a means of assessing the uncertainty associated with the KM-SUB model fit when extrapolating the model to environmental conditions outside the calibration range (Berkemeier et al., 2021). For both purposes, the number of model fits in the ensemble must be sufficiently large to fully grasp the remaining
model flexibility. The process of determining such a large set of fits can be computationally expensive. A surrogate model can either fully replace the KM, or assist in the fitting process by suggesting sampling points.

In this study, we evaluate the benefits of surrogate model-supported sampling by comparing the distribution of KM-SUB output MSLE for three different sampling approaches within the parameter boundaries presented in Table 1.

– Random log-uniform sampling

– Metropolis Hastings algorithm (MHA)-directed sampling

– NN-suggested sampling

We choose an MSLE of 0.016 as sufficient agreement of model and experiment. For NN-suggested sampling, we perform a random log-uniform screening of the NN surrogate model in batches of 10,000 samples until we find 5000 "NN-suggested





fits" with MSLE < 0.016 and feed these pre-sampled parameter sets into KM-SUB. KM-SUB outputs with an MSLE below
0.016 we refer to as "fits".

As directed sampling approach, we apply the Metropolis Hastings algorithm (MHA), a common Markov chain Monte Carlo
method to sample multivariate distributions with high numbers of dimensions (Chib and Greenberg, 1995; Robert and Casella,
1999). We determine the maximum step size of the MHA by basic testing on smaller subsets and find that a step size of 0.1
is a good compromise between a high acceptance ratio and sufficient exploration of the entire parameter space. Step size is
here defined as the maximal parameter variation as fraction of the total logarithmic parameter space. For comparability of the
aggregate computational effort, each sampling is performed on an 11th Gen Intel(R) Core(TM) i5-1145G7 CPU with 2.6 GHz.

## 2.7 Hardware and software tools

Training data acquisition with KM-SUB was performed in Matlab on the high performance computing system Cobra at the
Max Planck Computing and Data Facility (MPCDF). Model training of the NN was performed in Python (V.3.6) using the
packages Keras (2.3.0), TensorFlow (1.14.0), scikit-learn (0.22.1), NumPy (1.18.1), and pandas (0.25.3). Each model training
was conducted on one NVIDIA GeForce GTX 1080 Ti on the high performance computing cluster Mogon of the Johannes
Gutenberg University Mainz. For the PCE and sensitivity analysis, we use the Matlab-based software UQLab 1.3 (Marelli
and Sudret, 2014), which provides a framework for surrogate modelling and uncertainty quantification. We performed PCE
calculations on ETH Zurich's high performance computing cluster Euler, using 4 CPU per PCE calculation and up to 45 GB
of memory for the largest sample size (20,000).

To determine training times of the NN and PCE models, the required time for sample loading and file writing is disregarded
and only the true training time reported. For the PCE method, the time to reach 90 %, 50 % and 10 % of the initial amount of
Y, $N_{Y,0}$, is calculated by three separate models and training times are added to yield a combined training time for each training
sample size. For the NN method, one model can be set to return multiple values as output, thus, a single model is used for each
data set to predict all three output values collectively.

## 3 Results and discussion

### 3.1 Surrogate model training, accuracy and speed

Table 2 displays the test set errors and training times of surrogate models with the NN and PCE methods as a function of
training data set size. The best surrogate models achieve mean square errors (MSE) for logarithmic reaction times of 0.0049
for the NN method and 0.0137 for the PCE method. This corresponds to correlation coefficients $R^2$ of 0.995 and 0.991,
respectively. Figure 2 shows that these optimal versions for both surrogate models track the test data set remarkably well. The
MSE of test predictions is very similar between both approaches for the same training data set size. Error variance of the five
cross-validation NN models for the unseen test data is very low at $2.98 \times 10^{-6}$, indicating little to no over-fitting.





**Table 2.** Training times of surrogate models with the NN and PCE method.

| Training data set size | MSE of NN test predictions | NN training time (s) | MSE of PCE test predictions | PCE training time (s) |
|---|---|---|---|---|
| 50 | 1.03 | 2 | 1.44 | 3 |
| 100 | 0.718 | 2 | 0.328 | 3 |
| 200 | 0.398 | 3 | 0.313 | 4 |
| 500 | 0.172 | 7 | 0.196 | 5 |
| 1000 | 0.144 | 14 | 0.132 | 20 |
| 2000 | 0.104 | 28 | 0.078 | 144 |
| 5000 | 0.049 | 102 | 0.039 | 4232 |
| $1\times10^4$ | 0.025 | 67 | 0.022 | $3.28\times10^4$ |
| $2\times10^4$ | 0.014 | 260 | 0.014 | $2.17\times10^5$ |
| $5\times10^4$ | 0.010 | 326 | | |
| $1\times10^5$ | $8.6\times10^{-3}$ | 657 | | |
| $2\times10^5$ | $6.7\times10^{-3}$ | 961 | | |
| $5\times10^5$ | $4.9\times10^{-3}$ | 3250 | | |
| $1\times10^6$ | $6.6\times10^{-3}$ | 4097 | | |
| $2\times10^6$ | $7.3\times10^{-3}$ | 6477 | | |
| $4.3\times10^6$ | $5.9\times10^{-3}$ | $1.64\times10^4$ | | |

For data set sizes above 2000, the PCE model requires much more training time than the NN model. However, note that these training times of individual NN models disregard the necessity of hyperparameter tuning. While hyperparameter tuning is not required in an already established application, the total computation times of NN surrogate model training and hyperparameter tuning can be two orders of magnitude larger, depending on the extent of hyperparameter tuning that is performed. Hence, the use of a NN method is advisable when a large amount of training data is easily available and model accuracy is of high importance.

The PCE method on the other hand is limited in training data set size ($\leq$ 20,000) through calculation time and memory requirements in MATLAB. The PCE method is thus a good choice if the training data set is small or its acquisition is time-limiting, and when time-consuming hyperparameter tuning is not desired.

Both surrogate models calculate new output data orders of magnitude faster than the full model KM-SUB. The computation time of KM-SUB lies on the order of a few seconds per model run, while both, PCE and NN method, can generate large arrays of 10,000 individual surrogate model solutions in under one second.

### 3.2 Prediction of chemical loss and half-life

Fig. 3 visualizes the accuracy of the surrogate models (training set sizes 20,000 for PCE and 500,000 for NN) by generating five concentration-time-curves from various input parameter combinations and comparing to the full KM-SUB model. Input




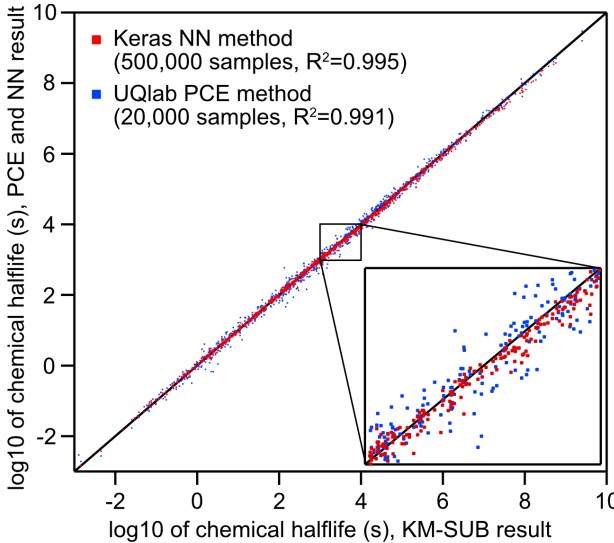

**Figure 2.** Comparison of the two surrogate models predicting the chemical half-life for heterogeneous chemistry on aerosol particles for a wide range of KM-SUB output (N=1000, test data set not part of training data set). The surrogate models were trained on 20,000 (PCE) and 500,000 (NN) KM-SUB data samples, respectively. Training times of models with this complexity fall below an upper feasibility range on a personal computer within few days of time. The inset shows a magnified section and spans from chemical half-lives of $10^3$ s ($\approx$ 15 min) to $10^4$ s ($\approx$ 3 h), a common range for laboratory experiments.

parameter sets were arbitrarily selected from the test set so that the results are spaced out homogeneously across KM-SUB

chemical half-lives. We see that over the wide range, both surrogate models closely represent the KM output, with the NN slightly outperforming the PCE method as result of the larger training set size.

Note that both methods are able to produce relatively good surrogate models (MSE $\approx$ 0.1) from only 1000 training data samples (Table 2), which depending on the user's application may already be accurate enough. We conclude that KM-SUB is a rather well-behaved model and suitable for these surrogate modelling techniques.

**3.3 Global sensitivity analysis with surrogate models**

An advantage of using a PCE surrogate model is that the Sobol' sensitivity indices can be extracted analytically (Sudret, 2008). We present the global sensitivity analysis for the 50 % lifetime (i.e. the chemical half-life) PCE model in Fig. 4. We can differentiate between first-order effects of a model input parameter, wherein the parameter alone influences the output, and interaction effects, wherein combinations of parameter values influence the output. In Fig. 4, first-order effects dominate the

total effect, accounting for 88 % of the model variance. Using the total Sobol' indices ($S^T$) as a metric, we can assess the overall influence of individual model parameters on the uncertainty of the model output. The input parameters with the largest influence on the chemical half-life of Y are the initial gas phase concentration of X ($[X]_{g,0}$, $S^T = 0.36$) and the radius of the particle ($r_p$, $S^T = 0.22$). Certain parameters have a very low influence ($S^T \leq 0.05$) on the chemical half-life, including the




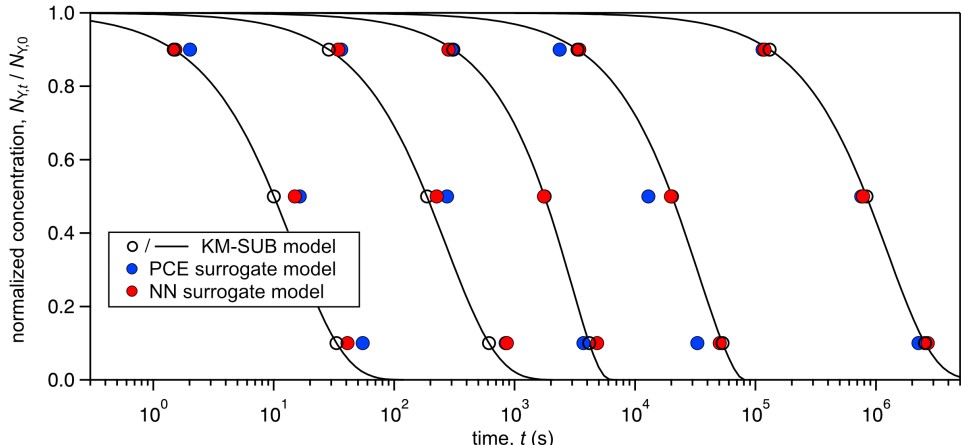

**Figure 3.** Comparison of time-dependent output of the surrogate models (PCE, blue markers; NN, red markers) with KM-SUB model output (black solid lines) for five arbitrarily chosen KM-SUB runs spanning seconds to weeks of reaction time. The surrogate models predicted time for depletion of 10, 50 and 90 % of reactant Y in the aerosol phase. KM-SUB output at these three stages is highlighted with black open markers.

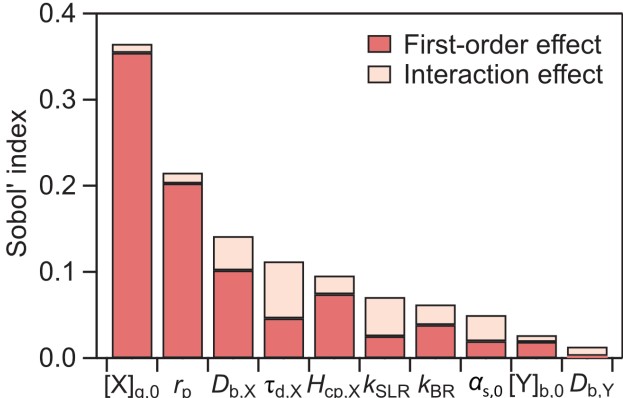

**Figure 4.** Results of global sensitivity analysis showing Sobol' sensitivity indices for the chemical half-life PCE model.

accommodation coefficient ($\alpha_{s,0,X}$), the initial concentration of Y ($[Y]_{b,0}$), and the bulk diffusion coefficient of Y ($D_{b,Y}$). This
means that variations in these parameters will in many cases not have a large effect on the chemical half-life, indicating that it will be difficult to constrain these parameters with measurements. Sensitivity analysis is thus a useful tool to understand model behavior and identify parameters which have the largest influence on model output.

It has to be noted that a low global sensitivity across the entire input parameter space does exclude the possibility that pockets in the parameter space exist where either of these parameters are very influential. Constraining the input parameter
space to smaller subsets can constrain the model to special kinetic regimes or limiting cases that exhibit characteristic profiles of parameter sensitivity (Berkemeier et al., 2013).





In most laboratory experiments, the particle radius and the initial concentration of X are known values. By fixing these parameters in the sensitivity analysis, a substantial fraction of the model variance is eliminated and other unknown parameters account for a more significant fraction of the overall model variance. To demonstrate how the importance of parameters varies
over different experimental conditions, we conducted sensitivity analyses by sampling the PCE surrogate model for specified values of $[X]_{g,0}$ and $r_p$ (Fig. 5a). Certain input parameters are consistently important across the range of experimental conditions, e.g. oxidant diffusivity ($D_{b,X}$) and solubility ($H_{cp,X}$). Other parameters, including $k_{BR}$ and $\tau_{d,X}$, have varying influences depending on the experimental conditions. For example, at a high $[X]_{g,0}$ and for large $r_p$, the total Sobol' index of $\tau_{d,X}$ is 0.14. Accordingly, the upper panel of Fig. 5b shows that the chemical half-life of Y only decreases slightly with increasing $\tau_{d,X}$. In
contrast, at low $[X]_{g,0}$ and for small $r_p$, the total Sobol' index increases to 0.31. In the lower panel of Fig. 5b, the chemical half-life of Y shows a stronger dependence on $\tau_{d,X}$. This can be understood because for small particles surface processes are more important and the surface concentration of X depends on its lifetime for desorption, especially at low gas phase concentrations. This information could be potentially useful for an experimental researcher, as it shows that experiments at low $[X]_{g,0}$ and small $r_p$ could be more helpful to constrain $\tau_{d,X}$ than experiments under other experimental conditions.
These calculations would have been very time consuming when carried out with the full KM. Hence, the combination of surrogate modelling and sensitivity analysis are a helpful, yet underutilized tool to design experiments that are best suited to constrain certain model parameters.

### 3.4   NN-supported global optimization

Utilizing the NN surrogate model, we illustrate the accelerated acquisition of parameter sets associated with KM-SUB outputs
in good agreement with experimental data, which is the key step in inverse modelling and optimization approaches. While uncertainty is introduced by surrogate models, their predictions can be obtained orders of magnitude faster than regular KM-SUB calculations. The uncertainty introduced by the NN can be minimized by additional sampling of a much smaller number of parameter sets with the KM. Re-sampling of NN-suggested solutions with the KM can avoid collection of false-positive fits (i.e. meeting the conditions for a "fit" in the NN model, but not in KM-SUB) and sampling in close vicinity of NN-suggested
solutions might avoid false-negative fits (i.e. not meeting the conditions for a "fit" in the NN model, but in KM-SUB).

We perform random parameter sampling in log-uniform space using the boundaries presented in Table 1 and find about 5000 NN-suggested fits in $1.84 \times 10^7$ parameter sets (0.027 % acceptance), requiring a total of 13,847 s (<4 hours). A comparable calculation with KM-SUB would take years on a desktop computer or days on a supercomputer. In contrast, re-sampling of the NN-suggested fits with KM-SUB to avoid false-positive fits is time-consuming, but feasible. The time required for sampling
of 5000 kinetic parameter sets (i.e. 5000×6 runs in KM-SUB) on a desktop computer ranges from 51,646 s ($\approx$ 14 hours) for NN-suggested sampling to 103,530 s ($\approx$ 29 hours) for random log-uniform sampling. The differences may be a result of the fraction of parameter sets where differential equation calculations of the KM require a very long time to terminate. They are often associated with very long reaction times and thus with large MSLE.

Fig. 6 shows the distributions of KM-SUB output MSLE for three different sampling methods: loguniform random sampling,
MHA-directed sampling, and NN-suggested sampling (Sect. 2.6). The NN-suggested sampling method greatly outperforms





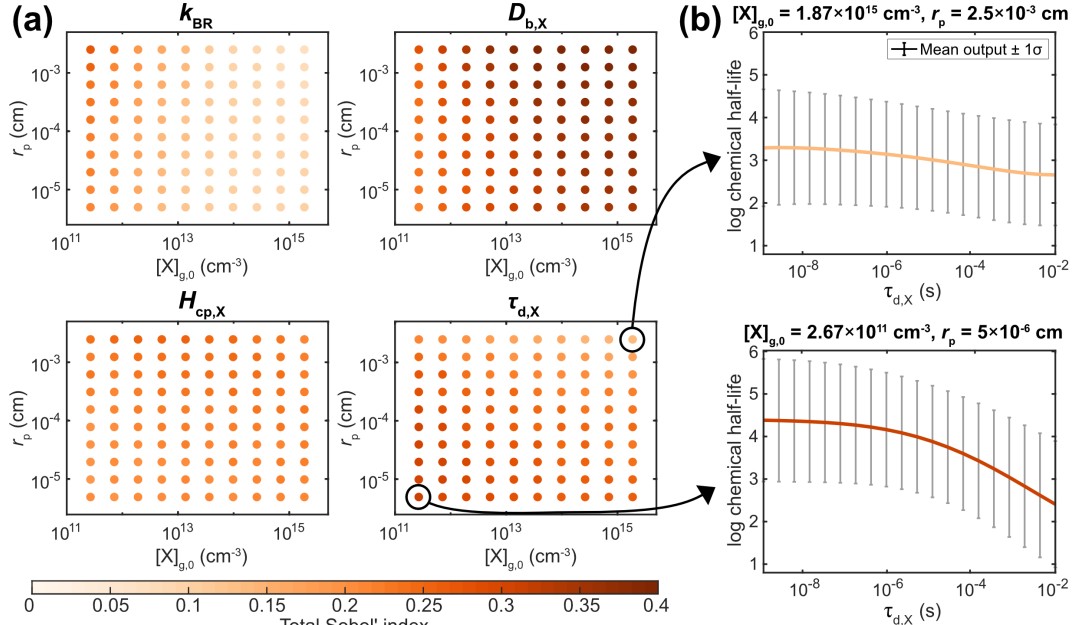

**Figure 5.** Detailed sensitivity analysis with the PCE method as a function of experimental conditions, i.e. the gas-phase concentration of X ($[X]_{g,0}$) and particle radius ($r_p$) (a) Total Sobol' indices of four KM input parameters: bulk reaction rate coefficient of X and Y ($k_{BR}$), bulk diffusion coefficient of X ($D_{b,X}$), solubility coefficient of X ($H_{cp,X}$), and desorption lifetime of X ($\tau_{d,X}$). (b) Relationship between the value of $\tau_{d,X}$ and the chemical half-life of Y for two selected experimental conditions.

both random and MHA-directed sampling. The number (fraction) of KM-SUB outputs with an MSLE < 0.016 is 1602 (32.04 %) for NN-suggested sampling, 21 (0.42 %) for directed KM-SUB-sampling, and 3 (0.06 %) for random sampling.

Fig. 7 compares the fitting parameter space of 5000 fits obtained with KM-SUB (panel a) and the NN surrogate model (panel b), exemplary for four kinetic parameters in a so-called scatter plot matrix. The off-diagonal elements in each matrix show bivariate scatter plots (top right) or densities plots (bottom left) depicting the relationship of two kinetic parameters within the fit ensemble. The diagonal elements are histograms showing frequency distributions of the individual parameters. The two scatter plot matrices show a clear resemblance of the fit parameter spaces between the surrogate model and the original KM. Much like the scatter plots of the original model fits, the scatter plots of the surrogate model fits can be used to identify areas that will not produce a fit to experimental data. For example, there are no fits with a slow surface reaction rate coefficient ($k_{SLR}$) and a high oxidant solubility ($H_{cp,X}$). However, some features in the scatter plots of the surrogate model deviate from those in the scatter plots of the original KM. We can visually identify areas in the scatter plots that indicate false-positive fits, i.e. being only occupied in the plots for the surrogate model. An absence of density in other areas, compared to the plots for the original model, suggests the existence of false-negative fits.

Whether it is worthwhile to train a surrogate model for a given optimization task depends strongly on the complexity of the KM and the difficulty of the optimization problem. For every application, there is a "break-even point" where the



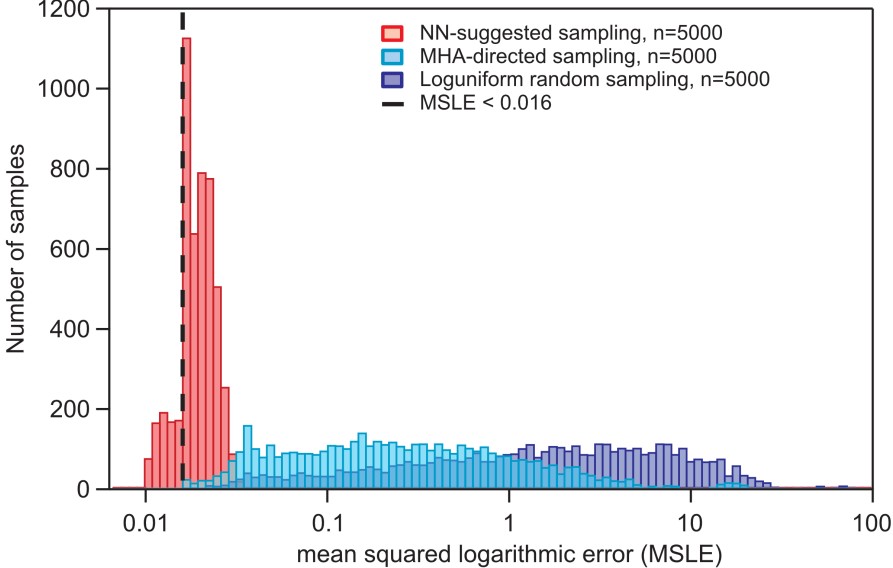

**Figure 6.** Distribution of KM-SUB output MSLE for three different sampling methods in comparison with six sets of experimental data, as described in Sec. 2.6. The dashed vertical line represents the threshold used for the acquisition of NN-suggested fits (MSLE < 0.016). The maximum step size for the MHA-directed sampling is 0.1.

computational expense of training a surrogate model is compensated by the acceleration of the optimization task(s). In this study, the computational effort required to obtain the training data for the best-performing surrogate model (500,000 KM-SUB sample runs) would only find ~350 fits if we had directed this initial sampling effort into fit acquisition using only KM-SUB. This is due to very low fraction of fits (0.42 %) without the aid of surrogate models and because KM-SUB has to be evaluated
six times, once for each laboratory data set. Thus, if the uniqueness of an optimization results must be determined, large amounts of laboratory data are available, or simply, if global optimization of the same model is required on a regular basis, training of a surrogate model for this task quickly becomes worthwhile.

## 4     Conclusions

In this study, we illustrate the application of artificial neural networks (NN) and polynomial chaos expansion (PCE) to generate
fast surrogate models for computationally expensive kinetic models (KM). As template KM, we use the kinetic multi-layer model of aerosol surface and bulk chemistry (KM-SUB, (Shiraiwa et al., 2010)), but the presented methods can equally be applied to other process models. Our findings suggest that after an initial investment of computational effort for training data sampling and model training, both methods yield models with very good correlations to KM-SUB outputs ($R^2 > 0.99$). Furthermore, we provide examples for the application of such surrogate models for inverse modelling and kinetic parameter
optimization: global sensitivity analysis with the PCE method and acceleration of global optimization with the NN. The results





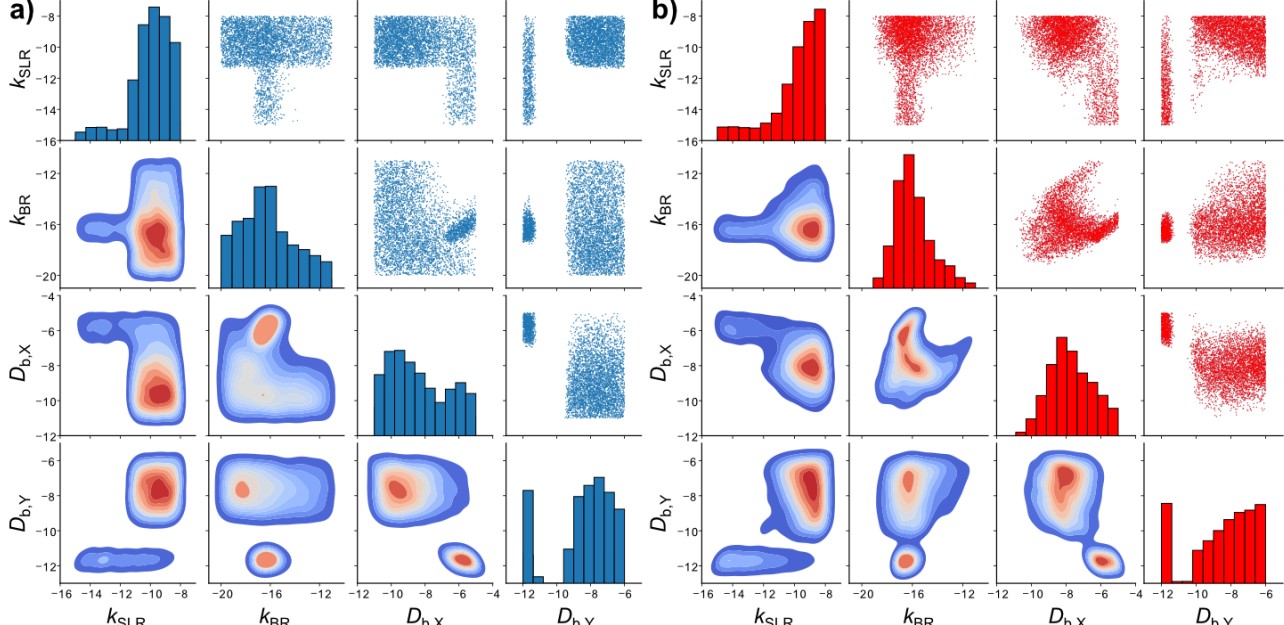

**Figure 7.** Scatter plot matrices of the fitting parameter space of 5000 fits to six experimental data sets of the ozonolysis of oleic acid aerosol (Appendix B) obtained with (a) KM-SUB and (b) the NN surrogate model. Shown are four out of seven optimized kinetic parameters. The diagonal elements are histograms showing the distributions of the individual fit parameter densities. The off-diagonal elements are scatter plots (top right) or densities (bottom left) of solutions for all possible combinations of two kinetic parameters. The KM-SUB fit ensemble originates from the application of the MHA with a step-size of 0.1 and the NN fit ensemble from loguniform random sampling.

indicate that surrogate models can aid in costly optimization tasks or help to select environmental system parameters for experiments that significantly constrain KM solution space, and thus global fit uncertainty.

It is important to note that errors of surrogate models are not simply based on a random deviation of surrogate model predictions from the values of the original KM, but on a divergence of the predicted parameter hyper-surface in specific areas,
for instance where training data is sparse. False-positive fits, i.e. parameter sets with associated surrogate model predictions in better agreement with experimental data as the delineated KM output, can simply be eliminated by re-sampling the parameter sets in question with the KM (Fig. 6). On the other hand, false-negative fits and their implications for inverse modelling approaches are much more difficult to address. While optimization hyper-surfaces can be scanned relatively quickly with a surrogate model, this is not the case for the much slower KM. Scatter plot matrices of the fitting parameter space are a valid
means of identifying areas that are occupied by false-negative fits, but a proper comparison (Fig. 7) requires computationally costly sampling with the KM.

Another potential application of surrogate models for KM is their utilization as modules in large-scale chemical transport models. As such models often require many calls of the respective module, direct use of models such as KM-SUB, where calculation time is on the order of seconds, is not feasible. Trained, predictive surrogate models, however, can easily be integrated



in existing modelling programs. This potentially allows the coupling of small-scale kinetic process models with large-scale chemical transport models for the simulation of weather, pollution, and climate. Kelp et al. (2022) recently demonstrated acceleration of a global model with an online-learned NN as chemistry module. The machine learning models presented in this study could be embedded in existing FORTRAN code in similar fashion.

*Code and data availability.* All training data as well as the source code used for obtaining NN and PCE models is archived on Zenodo
(https://doi.org/10.5281/zenodo.7214880; Berkemeier et al., 2022).

## Appendix A: Neural networks

### A1 Neural network architecture

A multilayer perceptron (MLP) represents a complex, non-linear function that maps an input to an output vector. Each individual node in a MLP represents a non-linear function, mapping from the sum of its inputs to an output, which is passed to the
following interconnected nodes. Connections between nodes are associated with weights that are optimized during training, in order to reduce model output error in comparison with the data set values. For this purpose, an optimization algorithm is used to minimize a previously defined loss function based on the final model output. In their entirety, these weights determine the output of the MLP based on a specific input and their adaptation based on the training data represent the learning process. The following equations show the principal mathematical functionality of neurons in a MLP, as elaborated in Kröse and van der
Smagt (1996):

$$s_k(t) = \sum_j w_{jk}(t)y_j(t) + \Theta_k(t) \tag{A1}$$

where $s_k(t)$ is the effective input of a neuron $k$ at time $t$, $w_{jk}$ the weight between neuron $j$ and $k$ and $y_j(t)$ the activation of the previous neuron $j$. This equation represents the input of a single computational node in the NN, which is based on the activation of connected previous nodes and the associated (trained or initialized) weights. $\Theta_k(t)$ represents an offset term. Of
this so called propagation rule, different adaptations have been proposed (Feldman and Ballard, 1982).

$$y_k(t+1) = F_k(y_k(t), s_k(t))) \tag{A2}$$

This equation introduces the activation function of neuron $k$ ($F_k$) that maps the neuron input $s_k(t)$ and the current activation $y_k(t)$ of the neuron to a new activation value. A common type of the activation function is a sigmoid-like function, as shown in the following equation:

$$y_k = F(s_k) = \frac{1}{1 + e^{-s_k}} \tag{A3}$$

The definition of input and activation functions of neurons determine the output of any NN, given a specific input and a set of weights. NN model training or learning describes the process of iterative modification of weights in order to shift the





output in a desired way. In most cases, this desired shift is a reduction of error towards the associated predictable values in the underlying population associated with the training data. If the model is well fitted to the training data but predicts further data

of the same population with much larger error, it is called over-fitted. Over-fitting describes overall ill generalization of a NN model. A common learning rule for nodes, the so called perceptron learning rule is shown in the following equation:

$$w_i(t+1) = w_i(t) + \Delta w_i(t) \tag{A4}$$

In order to adjust the weights, the output of the NN is compared with the associated training data values. If the prediction is inaccurate, the modification $\Delta \mathrm{w}_i$ is applied. For this iterative adjustment to be target-oriented, an optimizer is necessary to

reduce prediction error of the NN during training. Different optimizers are commonly used in machine learning applications, such as simple gradient methods like Stochastic Gradient Descent, where an estimate of the gradient (the direction of steepest descent) along with a selected step-size determines the variation of input parameters in the current step. As information in a feedforward NN, like a MLP, is only passed in one direction, a method called back-propagation is used to determine the direction and amount of weight adjustment in previous NN layers based on the error of the final prediction. More in-depth

explanations, definitions and examples for back-propagation and optimization throughout the learning process can be found in Rumelhart et al. (1995) and Hecht-Nielsen (1992), for further information regarding MLP and NN in general, see Almeida (2001) or Popescu et al. (2009).

## A2 Hyperparameter tuning

Comprehensive hyperparameter tuning is conducted every time a surrogate model is trained on different training data. In this

study, we focus on the investigation of data set sizes and training times. For this reason, and because our application of NN is not very common and only few information regarding successful model architectures and hyperparameters are available, only basic, plain network architectures are tested: MLP with up to five fully connected hidden layers and up to 4096 neurons in each of the layers. We perform hyperparameter tuning in three steps, aiming for an optimization of number of layers, layer activation functions, learning rate and batch size in the first, number of neurons in each layer in the second, and dropout rate

in the third step. For each step, we apply an adapted grid search where multiple well-performing hyperparameter sets from the previous step are extended by variation of the additionally optimized hyperparameter of the current step.

We performed relatively comprehensive hyperparameter tuning with 60 to 120 hyperparameter sets for each data subset, each tested set resulting in five models for the individual cross-validation folds. Sets of hyperparameters that lead to well-performing models can to some extent be adopted for approaches with similar preconditions regarding the number of in- and outputs or

training data set size. For a similar approach, we recommend a basic hyperparameter tuning with at least ten hyperparameter sets and 5-fold cross-validation. Best models are selected by average test set error of the five models for each of the cross-validation folds, using the mean squared error. The ranges of hyperparameters tested in this study are listed in Table A2 along with the hyperparameter values of the best performing models for large data sets.

Besides NN from the Keras package, other deep learning algorithms tested for this study are Random Forest Regressor,

Decision Tree Regressor, SGD Regressor, Ridge Regressor, Lasso, Logistic Regression, and MLP Regressor, provided by the



Python-library scikit-learn (Pedregosa et al., 2011). As most of the tested algorithms did not perform very well in basic tests, we focus on Keras as common and versatile tool for neural network application.

**Table A1.** Employed polynomial degree of the three PCE models (90, 50, and 10 % lifetime) as function of training data set size.

| Data set size | PCE 90 % $N_{Y,0}$ | PCE 50 % $N_{Y,0}$ | PCE 10 % $N_{Y,0}$ |
|---|---|---|---|
| 50 | 3 | 3 | 3 |
| 100 | 2 | 2 | 2 |
| 200 | 3 | 3 | 3 |
| 500 | 3 | 3 | 3 |
| 1000 | 4 | 4 | 4 |
| 2000 | 5 | 5 | 5 |
| 5000 | 7 | 6 | 6 |
| 10000 | 7 | 7 | 7 |
| 20000 | 7 | 7 | 7 |





**Table A2.** Descriptions and tested ranges for neural network hyperparameters used in the Python package Keras, as well as the recommendation based on our best-performing model.

| Parameter | Lower boundary | Upper boundary | Recommended value | Description |
|---|---|---|---|---|
| Number of hidden layers (HL) | 1 | 5 | 2 | The number of hidden layers in the NN - determines network size and strongly impacts computational cost |
| Activation functions[1] | "relu", "elu" or "sigmoid" | | All "relu" | Activation function for the neurons in each of the hidden layers |
| Number of neurons[1] | 4 | 4096 | (4096, 4096) | Also determines NN model size - large numbers are associated to increased computational coast and risk of over-fitting |
| Dropout rate[1] | 0.1 | 0.9 | 0.5 | The model ignores this fraction of all weights in this HL during training[2] |
| Optimizer | "Adam", "Nadam", "SGD" or "RMSprop" | | "Adam" | Optimizer for training process |
| Batch size | 4 | 128 | 16, depends on learning rate[3] | The number of training samples handled by model in a "batch" |
| Epochs | 4 | 60 | 32, until model loss converges | Number of training epochs |
| Learning rate | $10^{-5}$ | $10^{-1}$ | 0.0001 | Extent of variation of weights in attempt to decrease error |
| Decay | 0 | 0.9 | 0 | Decrease of learning rate throughout training epochs |

[1] Must be set for each individual HL

[2] A random fraction of weights obtained in previous training, determined in size by this parameter, is not considered during the current training. This "handicap"/restriction ensures, that the model is not capable of just "saving"/learning all the in- and associated outputs in the training data set throughout multiple training epochs (as this would be over-fitting).

[3] A larger batch size decreases training time and requires higher learning rates.



**Appendix B: Oleic acid ozonolysis data sets**

In Sect. 3.4, KM-SUB and the NN surrogate model are applied to six experimental data sets of the ozonolysis of oleic acid
aerosol available in the literature (Hearn and Smith, 2004; Ziemann, 2005; Gallimore et al., 2017; Berkemeier et al., 2021).
These data sets comprise flow tube, environmental chamber, and single-particle levitation techniques and are a subset of
data investigated earlier by Berkemeier et al. (2021), omitting the studies that investigated particles with a sodium chloride
core or in which the particle size was not measured. The experimental data sets are converted to normalized concentrations
($N_{Y,t}/N_{Y,0}$) and further simplified by fitting a mono-exponential decay ($A + B \cdot \exp(-\tau_e \cdot t)$) and evaluating the reaction time
at which 10, 50, and 90 % of oleic acid are consumed. Table B1 shows the environmental parameters (particle radius $r_p$,
ozone concentration $[X]_{g,0}$, and initial oleic acid concentration $[Y]_{b,0}$), the derived reaction times, and the mono-exponential
fit parameters. The remaining seven KM-SUB input parameters listed in Table 1 are optimized. Fig. B2 shows all data sets
alongside a fit ensemble of 50 KM-SUB fits with a fit correlation MSLE less than 0.016.

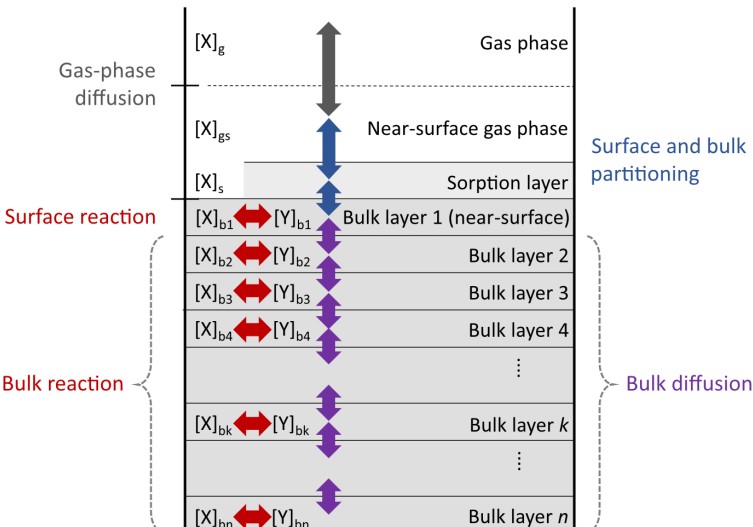

**Figure B1.** Compartments and processes of the kinetic multi-layer model of aerosol surface and bulk chemistry (KM-SUB).





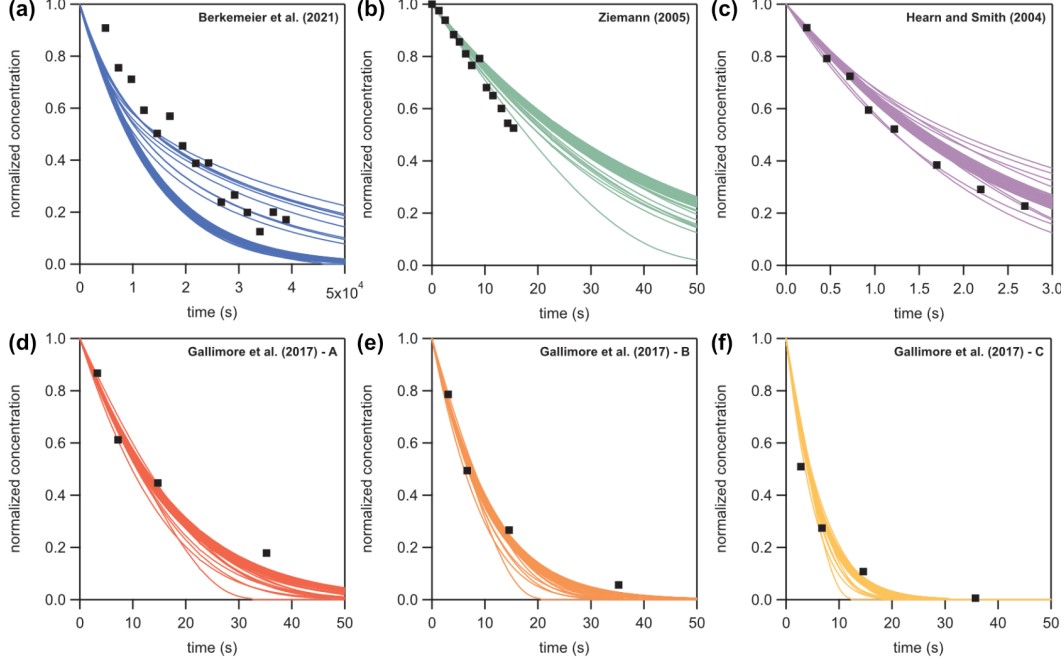

**Figure B2.** Fit ensembles of KM-SUB (N=50, colored lines) with MSLE<0.016 to six literature data sets (black square markers) of oleic acid aerosol ozonolysis displayed as normalized oleic acid concentrations ($N_{\mathrm{Y},t}/N_{\mathrm{Y},0}$).

**Table B1.** Model parameters for the global optimization of six oleic acid ozonolysis data sets.

| Data set | $r_\mathrm{p}$ (cm) | $[\mathrm{X}]_{\mathrm{g},0}$ (cm$^{-3}$) | $[\mathrm{Y}]_{\mathrm{b},0}$ (cm$^{-3}$) | $t_{10\%}$ (s) | $t_{50\%}$ (s) | $t_{90\%}$ (s) | $A$ | $B$ | $\tau_\mathrm{e}$ |
|---|---|---|---|---|---|---|---|---|---|
| Berkemeier et al. (2021) | $1 \times 10^{-3}$ | $1 \times 10^{13}$ | $1.89 \times 10^{21}$ | 24166 | 15892 | 52791 | 0 | 1 | $4.36 \times 10^{-5}$ |
| Ziemann (2005) | $2 \times 10^{-5}$ | $7 \times 10^{13}$ | $1.2 \times 10^{21}$ | 2.85 | 18.8 | † | 0 | 1 | $3.69 \times 10^{-2}$ |
| Hearn and Smith (2004) | $4 \times 10^{-5}$ | $2.5 \times 10^{15}$ | $1.89 \times 10^{21}$ | 0.196 | 1.29 | 4.28 | 0 | 1 | 0.538 |
| Gallimore et al. (2017) - A | $2.5 \times 10^{-5}$ | $2 \times 10^{14}$ | $1.89 \times 10^{21}$ | 1.91 | 12.6 | 41.7 | 0 | 1 | $5.52 \times 10^{-2}$ |
| Gallimore et al. (2017) - B | $2.5 \times 10^{-5}$ | $3.25 \times 10^{14}$ | $1.89 \times 10^{21}$ | 1.12 | 7.39 | 24.6 | 0 | 1 | $9.38 \times 10^{-2}$ |
| Gallimore et al. (2017) - C | $2.5 \times 10^{-5}$ | $5.51 \times 10^{14}$ | $1.89 \times 10^{21}$ | 11.2 | 3.37 | 0.512 | 0 | 1 | $2.06 \times 10^{-1}$ |

†: Too far outside data range.



**Abbreviations**

KM - Kinetic multi-layer model

KM-SUB - Kinetic multi-layer model of aerosol surface and bulk chemistry

MHA - Metropolis Hastings algorithm

MLP - Multilayer perceptron

MSE - Mean square error

MSLE - Mean squared (absolute) logarithmic error

NN - Neural network

PCE - Polynomial chaos expansion

*Author contributions.* TB and UK conceived the study. All authors designed research. TB (KM-SUB model), MK (NN model), AF and MM
(PCE model) wrote code and performed simulations. All authors discussed and interpreted calculation results. TB and MK led the writing of

the manuscript and overall design of graphics and tables. AF and MM co-led the writing and graphics for the sections applying PCE models.
All authors contributed to writing and editing.

*Competing interests.* The authors declare that they have no competing interests.

*Acknowledgements.* This work was funded by the Max Planck Society (MPG) and supported by ETH Zurich through ETH Research Grant
ETH-03 17-2. Aryeh Feinberg acknowledges financial support from ETH Zurich (ETH-39 15-2). The authors thank C. Mattei and J. Wilson

for helpful discussions. We thank P. Ziemann, G. D. Smith and P. Gallimore for providing published data in tabulated form. The authors grate-
fully acknowledge the computing time granted on the supercomputer Mogon at Johannes Gutenberg University Mainz (hpc.uni-mainz.de)
and on the supercomputer Cobra at the Max Planck Computing and Data Facility (mpcdf.mpg.de).



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
