# Peer review of "Accelerating models for multiphase chemical kinetics through machine learning with polynomial chaos expansion and neural networks"

_EGUsphere, 2022_

## Author Response (AR1)

**Response to Reviewer 1**

Berkemeier et al. present two surrogate model approaches, polynomial chaos expansion and neural networks, used to predict discrete reaction times of kinetic multi-layer models and used for sensitivity analysis and inverse modelling. The paper is well-organized, concise, and appropriate for GMD. I recommend its publication after addressing the comments below.

We would like to thank reviewer 1 for taking the time to review our manuscript and highly appreciate the positive and constructive feedback. We enumerated each of their comments (black) and address them in point-by-point answers (blue). Quotes from the revised manuscript are indented and given in italic font with changes highlighted in red. Line numbers refer to the revised manuscript with markup.

**R1-1** One general comment: It is not clear in a first read what the surrogate models are predicting (in other words, what the targets are). On a second read, it seems like the targets are the order of magnitudes of 3 different reaction times: the 90% reaction time, half-life, and 10% reaction time. Using these targets instead of concentration targets makes sense if the application is machine learning aided sampling rather than full model replacement. However, many readers (including myself) might initially assume that surrogate models replicate the output of their reference model, so this non-standard target should be addressed earlier on.

Thank you very much for this valuable input that was also shared by reviewer 2. First, we would like to highlight the paragraph in the original, submitted manuscript that addresses this question:

> l. 94-97 - *The outputs of KM-SUB are concentration profiles over space and time, but in this study, we summarized KM-SUB output as the total number of Y in a single aerosol particle at time t ($N_{Y,t}$). To minimize data storage requirements, we reduce the full KM-SUB time series to three output values, the time required to reach 90 %, 50 % (i.e. the chemical half-life), and 10 % of $N_{Y,0}$ by interpolation of primary model output.*

We agree that this point should be made clearer in the manuscript and revisit this idea now in two further instances in the revised manuscript and added a discussion of the choice to simplify KM-SUB model output.

> l. 209-210 - *Neural networks (NN) and polynomial chaos expansion (PCE) are used to emulate the reaction time of a multiphase chemical system in KM-SUB.*

> l. 316-321 - *To reduce data storage requirements in sampling and simplify emulation, the complex model output of KM-SUB, i.e. concentration profiles of all reactants over space and time, is reduced to the reaction time of the system to reach a certain reaction progress as this is a typical observable in labora-*

*tory experiments. We note that other derivatives of KM-SUB model output, such as the uptake coefficient of the reactant gas to the aerosol surface, could be chosen depending on the target application of the surrogate model. The choice will also depend on whether KM-SUB is used to parameterize a sub-grid process in the model or part of the chemical integrator. Emulation of the entire KM-SUB output may be feasible and could be facilitated by data compression methods such as autoencoders, singular value decomposition or principal component analysis [Bishop and Nasrabadi, 2006, Petersen et al., 2008, James et al., 2013, Kingma and Welling, 2013].*

**R1-2** The methodology would be easier to follow if the output and purpose of the surrogate models was described early on. This could be done on line 5 of the abstract, for example "we show how machine learning methods can generate inexpensive surrogate models \add{to predict reaction times for}\remove{based on} the kinetic multi-layer model of...", or something to an equivalent effect.

We thank the reviewer for this good suggestion. The sentence in the abstract now reads:

l. 4-6 - *In this study, we show how machine learning methods can generate inexpensive surrogate models for the kinetic multi-layer model of aerosol surface and bulk chemistry (KM-SUB) to predict reaction times in multiphase chemical systems.*

**R1-3** Not required, but a suggestion that might improve clarity: could the authors perhaps include a table for the 3 outputs, similar to Table 1 for the input parameters? This might help new readers understand the targets more easily.

After the changes in response to reviewer comment R1-1 and R1-2, we think it is not necessary to also include a table for this information.

**R1-4** There is existing recent work towards network surrogate models in aerosol applications for atmospheric chemistry and physics. Though I have no objection to the mention of applications outside the field, e.g. chemical engineering and materials science, the authors could include several relevant references to round out the discussion around line 60, which would help embed this work in recent literature on ML surrogate modelling for aerosol applications.

a. Surrogate model of MOSAIC, which includes heterogenous chemistry and mass transfer: https://doi.org/10.1029/2020JD032759

b. Mass-conserving surrogate model of mass transfer for bulk/surface partitioning: https://doi.org/10.5194/gmd-15-3417-2022

c. Physics-informed surrogate model of aerosol microphysics, specifically the M7 module: https://doi.org/10.1017/eds.2022.22.

Berkemeier et al. train surrogate models on a more granular kinetic multilayer model, for use in sampling rather than a solver / operator replacement (thus predicting a different output instead of concentration distributions). Inclusion of these more relevant references will allow the authors to make clear what the contribution of this paper is in the context of the field.

We would like to thank reviewer for bringing these references to our attention. We added the suggested references as well as others to the paragraph that addresses previous work on machine learning surrogate modelling.

> l. 63-68 - *Machine learning-based surrogate models have also found application as modules in geoscientific models, including large-scale atmospheric chemistry, transport and climate models, to reduce computational cost in very demanding tasks such as atmospheric convection [O'Gorman and Dwyer, 2018], gas-phase and heterogeneous chemistry [Keller and Evans, 2019, Kelp et al., 2020, Sturm and Wexler, 2022], or aerosol and cloud microphysics [Rasp et al., 2018, Harder et al., 2022]. These surrogate models function either as parameterizations for subgrid processes or replace the chemical integrator.*

**R1-5** Line 65: "In the PCE approach, the full model is represented as an infinite series..." I would get rid of the word infinite here, or perhaps say "In the PCE approach, the full model can be approximated by a series...". The PCE approach assumes the model can be represented as an infinite series, but is not applied in practice. This isn't clearly stated until mention of truncation around line 136.

We thank the reviewer for this helpful comment and removed the word 'infinite' in l. 70.

**R1-6** Figure 2: Are these correlation coefficients for just the log10 of half-life or for all reaction times, as line 205 suggests? Also, maybe "halflife" could be hyphenated to match the text? The inset for the laboratory-experiment relevant timescale is a nice addition.

We agree with the reviewer that this could be stated more clearly and added this information to the sentence in question:

> l. 213 -*Figure 2 shows that these optimal versions for both surrogate models track the chemical half-life in the test data set remarkably well.*

We also changed the x-axis label of Fig. 2 to "log10 of chemical half-life (s), KM-SUB result".

**Response to Reviewer 2**

Berkemeier et al. present work using neural networks and polynomial chaos expansion to emulate complex models of multiphase kinetics for atmospheric aerosols. They find that both techniques are suitable for emulation, and assess the benefits and drawbacks of each approach. In general, the work is well-presented, addresses an important topic, and provides a valuable contribution to the scientific literature. I have a few comments that I believe should be addressed before this paper is accepted for publication.

We would like to thank reviewer 2 for taking the time to review our manuscript and highly appreciate the positive and constructive feedback. We enumerated each of their comments (black) and address them in point-by-point answers (blue). Quotes from the revised manuscript are indented and given in italic font with changes highlighted in red. Line numbers refer to the revised manuscript with markup.

**R2-1** Motivation. The motivation of this work makes it challenging to understand exactly what the emulators are being used to emulate. This becomes clear on a more careful read, but I suggest the authors clarify for readability.

Thank you very much for this valuable input that was also shared by reviewer 1. Please refer to the response to R1-1 above.

**R2-2** Emulator Performance. The authors demonstrate very good bulk performance of their emulators in predicting the target quantity. However, the error of the methods as a function of the input parameter space is important to assess (e.g., error as a function of initial gas phase concentration). If the error is largely independent of the input parameter space, a sentence clarifying this would suffice.

We thank the reviewer for this good suggestion and added a figure to the supplementary material showing the error of the method as a function of each input parameter (Fig. 1). The absolute logarithmic error in chemical half-life was found to be largely independent of the value of input parameters.

We added the following sentence to the main text:

> l. 216-217 - *We found no significant correlations between surrogate model error and the values of the 10 model input parameters (Fig. S1).*

[Figure]

Figure 1: Absolute logarithmic error of the surrogate model for the 50 % reaction time (chemical half-life) in the test data set (N=1000) as a function of all 10 model input parameters. Due to the large size of the test data set and the variability in model error, errors are depicted as moving averages between the n=5 (blue dots) and n=100 (red lines) closest neighbours in a sorted list with ascending parameter values, respectively. No significant dependence of surrogate model error on parameter values is observed.

**R2-3** L54-56 Much work on NN in the atmospheric science has been done since 1998. I suggest improving the citations here.

Thank you very much for this valuable input that was also shared by reviewer 1. We have included a new paragraph addressing this issue as outlined in our response to reviewer comment R1-4.

**R2-4** L93 Were other normalization strategies explored (e.g., log-normal outputs)?

We are not certain what reviewer 2 refers to with "log-normal outputs". We simply took the decadic logarithm of the model output value and then uniformly transformed the output onto the interval [0,1].

**R2-5** L190 These packages need citations.

Thank you very much for pointing out this oversight. The packages are now referenced as follows:

> l. 194-196 - *Model training of the NN was performed in Python 3.6 using the packages Keras 2.3.0 [Chollet et al., 2015], TensorFlow 1.14.0 [Abadi et al., 2015], scikit-learn 0.22.1 [Pedregosa et al., 2011], NumPy 1.18.1 [Harris et al., 2020], and pandas 0.25.3 [McKinney et al., 2010].*

**R2-6** Table A2 Some of these hyperparameters selected are at the upper boundary of the sampled parameter space. This suggests that additional exploration would benefit model performance. Particularly for the shallow NN number of neurons parameter.

We would like to thank reviewer 2 for bringing this to our attention. Please note that the table summarizes hyperparameter values taken into account for the comprehensive grid-based hyperparameter tuning, while a multitude of initial tests has been conducted prior to this step. We found that larger numbers of neurons even for shallow NN led to over-fitting, indicated by a significantly lower training loss in comparison with the test error. Additionally, the memory requirements associated with "broader" NN models would have posed a limitation on the computer cluster. However, we fully agree that this relevant information should be provided in the manuscript. We added the following footnote to Table A2:

> Table A2: - [2] *Larger numbers of neurons per layer lead to over-fitting and, with the hardware setup in this study, memory limitations on the computational cluster.*

**References**

M. Abadi, A. Agarwal, P. Barham, E. Brevdo, Z. Chen, C. Citro, G. S. Corrado, A. Davis, J. Dean, M. Devin, S. Ghemawat, I. Goodfellow, A. Harp, G. Irving, M. Isard, Y. Jia, R. Jozefowicz, L. Kaiser, M. Kudlur, J. Levenberg, D. Mané, R. Monga, S. Moore, D. Murray, C. Olah, M. Schuster, J. Shlens, B. Steiner, I. Sutskever, K. Talwar, P. Tucker, V. Vanhoucke, V. Vasudevan, F. Viégas, O. Vinyals, P. Warden, M. Wattenberg, M. Wicke, Y. Yu, and X. Zheng. TensorFlow: Large-scale machine learning on heterogeneous systems, 2015. URL `https://www.tensorflow.org/`. Software available from tensorflow.org.

C. M. Bishop and N. M. Nasrabadi. *Pattern recognition and machine learning*, volume 4. Springer, 2006.

F. Chollet et al. Keras, 2015. URL `https://github.com/fchollet/keras`.

P. Harder, D. Watson-Parris, P. Stier, D. Strassel, N. R. Gauger, and J. Keuper. Physics-informed learning of aerosol microphysics. *Environ. Data Sci.*, 1:e20, 2022. ISSN 2634-4602. doi: 10.1017/eds.2022.22. URL `https://www.cambridge.org/core/product/identifier/S263446022200022X /type/journal_article`.

C. R. Harris, K. J. Millman, S. J. van der Walt, R. Gommers, P. Virtanen, D. Cournapeau, E. Wieser, J. Taylor, S. Berg, N. J. Smith, R. Kern, M. Picus, S. Hoyer, M. H. van Kerkwijk, M. Brett, A. Haldane, J. F. del Río, M. Wiebe, P. Peterson, P. Gérard-Marchant, K. Sheppard, T. Reddy, W. Weckesser, H. Abbasi, C. Gohlke, and T. E. Oliphant. Array programming with NumPy. *Nature*, 585(7825):357–362, Sept. 2020. doi: 10.1038/s41586-020-2649-2. URL `https://doi.org/10.1038/s41586-020-2649-2`.

G. James, D. Witten, T. Hastie, and R. Tibshirani. *An introduction to statistical learning*, volume 112. Springer, 2013.

C. A. Keller and M. J. Evans. Application of random forest regression to the calculation of gas-phase chemistry within the GEOS-Chem chemistry model v10. *Geosci. Model Dev.*, 12(3):1209–1225, Mar. 2019. ISSN 1991-9603. doi: 10.5194/gmd-12-1209-2019. URL `https://gmd.copernicus.org/articles/12/1209/2019/`.

M. M. Kelp, D. J. Jacob, J. N. Kutz, J. D. Marshall, and C. W. Tessum. Toward stable, general machine-learned models of the atmospheric chemical system. *J. Geophys. Res. Atmos.*, 125(23): e2020JD032759, 2020. doi: https://doi.org/10.1029/2020JD032759. URL `https://agupubs.onlinelibrary.wiley.com/doi/abs/10.1029/2020JD032759`. e2020JD032759 2020JD032759.

D. P. Kingma and M. Welling. Auto-encoding variational bayes. *arXiv preprint arXiv:1312.6114*, 2013.

W. McKinney et al. Data structures for statistical computing in python. In *Proceedings of the 9th Python in Science Conference*, volume 445, pages 51–56. Austin, TX, 2010.

P. A. O'Gorman and J. G. Dwyer. Using Machine Learning to Parameterize Moist Convection: Potential for Modeling of Climate, Climate Change, and Extreme Events. *J. Adv. Model. Earth Syst.*, 10(10):2548–2563, Oct. 2018. ISSN 1942-2466, 1942-2466. doi: 10.1029/2018MS001351. URL https://onlinelibrary.wiley.com/doi/10.1029/2018MS001351.

F. Pedregosa, G. Varoquaux, A. Gramfort, V. Michel, B. Thirion, O. Grisel, M. Blondel, P. Prettenhofer, R. Weiss, V. Dubourg, J. Vanderplas, A. Passos, D. Cournapeau, M. Brucher, M. Perrot, and E. Duchesnay. Scikit-learn: Machine learning in Python. *J. Mach. Learn. Res.*, 12:2825–2830, 2011.

K. B. Petersen, M. S. Pedersen, et al. *The matrix cookbook*, volume 7. Technical University of Denmark, 2008.

S. Rasp, M. S. Pritchard, and P. Gentine. Deep learning to represent subgrid processes in climate models. *Proc. Natl. Acad. Sci.*, 115(39):9684–9689, Sept. 2018. ISSN 0027-8424, 1091-6490. doi: 10.1073/pnas.1810286115. URL https://pnas.org/doi/full/10.1073/pnas.1810286115.

P. O. Sturm and A. S. Wexler. Conservation laws in a neural network architecture: enforcing the atom balance of a Julia-based photochemical model (v0.2.0). *Geosci. Model Dev.*, 15(8):3417–3431, Apr. 2022. ISSN 1991-9603. doi: 10.5194/gmd-15-3417-2022. URL https://gmd.copernicus.org/articles/15/3417/2022/.